Effects of shading stress on endogenous hormone levels in Eleutherococcus giraldii: hormonal dynamics and growth strategy analysis

Huang Xu Feng 1 2 3
Gu Rui 4 664893924@qq.com
Chen Guo Peng 4
Ding Rong 4
1 School of Pharmacy, Sichuan College of Traditional Chinese Medicine , Mianyang, Sichuan , China
2 Northwest Sichuan Laboratory of Traditional Chinese Medicine Resources Research and Development Utilization, Sichuan College of Traditional Chinese Medicine , Mianyang, Sichuan , China
3 Mianyang Key Laboratory of Development and Utilization of Chinese Medicine Resources, Sichuan College of Traditional Chinese Medicine , Mianyang, Sichuan , China
4 School of Ethnic Medicine, Chengdu University of Traditional Chinese Medicine , Chengdu, Sichuan , China
Serim Ahmet Tansel
Electronic publication date: 2025 Oct 1
Publication date: 2025
Volume: 13
Electronic Location ID: e20135
Received 2025 May 19; Accepted 2025 Sep 4
Copyright: © 2025 Huang et al.
Copyright year: 2025
Copyright holder: Huang et al.
License: This is an open access article distributed under the terms of the Creative Commons Attribution License, which permits unrestricted use, distribution, reproduction and adaptation in any medium and for any purpose provided that it is properly attributed. For attribution, the original author(s), title, publication source (PeerJ) and either DOI or URL of the article must be cited.
License URL: https://creativecommons.org/licenses/by/4.0/

Keywords: Eleutherococcus giraldii, Shade-induced stress, Endogenous hormones, Hormonal equilibrium

Funding: National Natural Science Foundation of China 81173476 This work was supported by the National Natural Science Foundation of China (No. 81173476). The funders had no role in study design, data collection and analysis, decision to publish, or preparation of the manuscript.

==============================
Background

Eleutherococcus giraldii (E. giraldii) is a quintessential traditional Chinese medicinal plant with significant developmental potential. Its growth and development are highly responsive to environmental factors, particularly light conditions. However, the endogenous hormonal changes in E. giraldii under shading stress remain unclear, and its adaptive growth strategies require further investigation.

Methods

Experimental groups with varying light transmittance (29.12%, 39.68%, and 100%) were established using shade nets, comprising moderate shading, light shading, and control groups. The endogenous hormone contents in apical and lateral leaves of E. giraldii were quantified using the enzyme-linked immunosorbent assay (ELISA). Statistical analysis and graphical presentations were performed using IBM SPSS Statistics 27 and Origin 2022 software.

Results

Shading treatment consistently promoted the accumulation of isopentenyl adenine nucleoside (iPA), zeatin riboside (ZR), gibberellic acid 3 (GA3), and abscisic acid (ABA) in both apical and lateral leaves of E. giraldii, while exhibiting differential effects on indoleacetic acid (IAA)-enhancing its content in lateral leaves but reducing it in apical leaves. Quantitative analysis revealed that moderate shading induced: (1) maximal increases in iPA, ZR, and GA3 levels in apical leaves (P < 0.05); (2) maximal increases in ABA, iPA, ZR, and GA3 concentrations as well as the ABA/ZR ratio in lateral leaves; (3) maximal reductions in the ABA/iPA and ABA/GA3 ratios in apical leaves along with the ABA/GA3 ratio in lateral leaves; (4) decreased ABA/IAA and ABA/ZR in apical leaves coupled with increased ABA/iPA in lateral leaves. Under light shading conditions, the most substantial changes included: (1) greatest ABA enhancement in apical leaves and IAA accumulation in lateral leaves; (2) most pronounced declines in IAA content (apical leaves) and ABA/IAA ratio (lateral leaves); (3) elevated ABA/IAA and ABA/ZR ratios in apical leaves with concurrent reduction of ABA/iPA in lateral leaves. Importantly, comprehensive correlation analysis demonstrated positive correlations among all examined hormones (ABA, IAA, iPA, ZR, and GA3) in both leaf types, indicating systemic hormonal coordination during shade adaptation.

Conclusion

Shading stress significantly restructured endogenous hormone profiles and their homeostasis in E. giraldii. Under moderate shading conditions, E. giraldii likely adopted a conservative strategy characterized by “apical leaf growth promotion coupled with lateral leaf growth restriction”, whereas mild shading induced an expansive strategy featuring “apical leaf growth inhibition coordinated with moderate lateral leaf expansion”. These findings provide a theoretical foundation for optimizing cultivation protocols and offer new insights into phytohormonal dynamics in shrubs under light limitation.

Introduction

The Eleutherococcus giraldii (E. giraldii) is a perennial shrub belonging to the Araliaceae family, widely distributed in Sichuan, Gansu, Ningxia, and other provinces of China (Jia & Zhang, 2016). Its dried stem bark is a classic traditional Chinese medicine included in the Sichuan Traditional Chinese Medicine Standards (2010 edition) and is known for its effects in dispelling wind-dampness, promoting joint mobility, and strengthening bones and tendons (Sichuan Provincial Food and Drug Administration, 2010). Modern research has demonstrated that E. giraldii bark exhibits anti-inflammatory, antiarrhythmic, hepatoprotective, antitumor, and antiviral properties (Li et al., 2019; Chen et al., 2018; Wang et al., 1992), indicating significant potential for further development.

Shading, a common light stress treatment, is widely used in studies on plant light adaptation mechanisms, shade tolerance evaluation, and cultivation management optimization. By reducing light intensity, shading can significantly influence physiological and biochemical processes in plants (Gong et al., 2022). Research indicates that shading treatment disrupts photosynthetic efficiency in green plants, alters the synthesis and distribution of endogenous hormones, and may trigger a series of related physiological responses (Li et al., 2023).

Enzyme-linked immunosorbent assay (ELISA), a well-established hormone detection technique, has been extensively validated for its specificity, sensitivity, and reliability through numerous experimental studies. Widely applied in plant physiology research, this method enables precise quantitative analysis of various plant endogenous hormones by leveraging the specific binding reaction between antigens and antibodies (Aydin et al., 2025).

Currently, research on the physiological and biochemical responses of E. giraldii remains limited, with particularly scarce reports regarding its endogenous hormone regulation. In this study, we investigated 5-year-old cultivated E. giraldii plants under shading treatments with varying light transmittance levels (29.12%, 39.68%, and 100%). Using ELISA, we quantitatively analyzed changes in abscisic acid (ABA), indoleacetic acid (IAA), isopentenyl adenine nucleoside (iPA), zeatin riboside (ZR), and gibberellic acid 3 (GA3) levels between experimental and control groups. Our findings elucidate the effects of shading stress on endogenous hormone dynamics in E. giraldii, providing theoretical support for its cultivation practices and contributing to broader understanding of hormonal regulation in shrubs under light-limiting conditions.

Materials and Methods

Study site description

The experimental site is located in Ruowo Village, Sanlong Township, Maoxian County (31°47′00.19″ N, 103°30′13.66″ E; altitude: 2,600 m asl), Aba Prefecture, Sichuan Province, China. The region exhibits distinct wet and dry seasons, with a mean annual temperature of 11.2 °C and ≥10 °C annual accumulated temperature of 3,293.3 °C. Annual precipitation averages 494.8 mm (>80% occurring from May to October), while annual evaporation reaches 1,332.4 mm. The south-facing slope (sunny aspect) features neutral loam soil with the following available nutrient levels: low phosphorus, moderate nitrogen, and high potassium (Chen et al., 2017). The site contains over 200 cultivated E. giraldii plants (5-year-old), providing sufficient experimental material (Fig. 1).

Figure 1 Five-year-old cultivated E. giraldii plants.

Methods

Experimental design

On June 30, 2023, eighteen healthy E. giraldii shrubs with uniform growth characteristics (plant height: 0.8–1.2 m; canopy width: 0.4–0.5 m) were randomly selected and assigned to three treatment groups (n = 6 per group) in a completely randomized design: (1) light shading (S1, 39.68% light transmittance), (2) moderate shading (S2, 29.12% light transmittance), and (3) control (CK, 100% light transmittance). The experimental groups (S1, S2) were subjected to shading treatment using commercially available polypropylene (PP) shade nets (pore size: 0.8–1.2 mm; light transmittance: 30–50%; Sichuan LvYin Shade Net Co., Ltd., Chengdu, China). Specifically, S1 was treated with single-layer shade nets while S2 received double-layer nets. To maintain proper air circulation, the upper edges of the nets were positioned 1 m above ground level with the lower edges fixed at 0.4 m height (Fig. 2). Light transmittance was measured in the afternoon under clear sky conditions using a TES-1339 portable illuminometer (TES Electrical Electronic Corp., Taipei, Taiwan).

Figure 2 Experimental shading design for E. giraldii.

The schematic diagram shows three key elements: (1) the dashed line A1 marking ground level, (2) lines A2 and A3 representing the shading net positions, and (3) red circles indicating stem nodes along the plant axis.

Sample collection and endogenous hormone assay

Sample collection: Apical and lateral leaves (~2 g each) from E. giraldii shrubs across treatment groups were collected on July 15 (7/15) and July 31 (7/31). Samples were weighed using an electronic balance (FA2004; MEIMIAN, Jinghai, China), labeled, immediately frozen in liquid nitrogen (−80 °C), and stored in polyethylene bags at −80 °C until analysis.

Hormone extraction: Frozen tissue (0.3–0.5 g) was homogenized in pre-chilled mortar with phosphate-buffered saline (PBS, pH 7.4, 1:9 w/v) under dim light. The homogenate was transferred to 10 mL centrifuge tubes, incubated at 4 °C for 3 h, and centrifuged (3,500 rpm, 10 min, 4 °C). Supernatants were purified using C18 solid-phase extraction columns (Waters, Milford, MA, USA).

ELISA quantification: Hormone concentrations (ABA, IAA, iPA, ZR, GA3) were determined using commercial ELISA kits (MEIMIAN, Jiangsu, China) following manufacturer protocols. Absorbance was measured at 450 nm (EMax® PLus Microplate Reader; Molecular Devices, Silicon Valley, CA, USA).

PBS preparation: 0.27 g potassium dihydrogen phosphate (KH2PO4), 1.42 g disodium hydrogen phosphate (Na2HPO4), 8 g sodium chloride (NaCl), and 0.2 g potassium chloride (KCl) were dissolved in 800 mL deionized water. The pH was adjusted to 7.2–7.4 with concentrated hydrochloric acid (HCl) before final volume adjustment to 1 L.

Data processing and analysis

The raw data were initially organized using Microsoft Office 2021 (primarily Microsoft Excel), followed by statistical analysis with IBM SPSS Statistics 27 (including Shapiro-Wilk normality test and one-way ANOVA). Graphs were generated using Origin 2022. Statistical results are presented as mean ± standard deviation (x̄ ± s), with different lowercase letters indicating significant differences between groups at P < 0.05.

Results

Endogenous hormone levels in apical leaves

Shading stress did not alter the increasing trends of ABA, IAA, iPA, ZR, and GA3 in the parietal leaves of E. giraldii, but affected their incremental changes (Fig. 3). On July 15 and July 31, ABA levels followed the order: light shading group > moderate shading group > control group (Fig. 3A), while iPA levels showed: moderate shading group > light shading group > control group (Fig. 3B). For IAA content on July 15, the ranking was light shading group > control group > moderate shading group, but reversed to control group > moderate shading group > light shading group by July 31 (Fig. 3C). ZR and GA3 levels on July 15 exhibited light shading group > moderate shading group > control group, but shifted to moderate shading group > light shading group > control group by July 31 (Figs. 3D, 3E). Moreover, on July 31st, significant differences were observed in apical leaf iPA and ZR levels between the moderate shading group and both the light shading and control groups (P < 0.05). Similarly, GA3 concentrations in apical leaves showed significant variations between the control group and both shading treatment groups (P < 0.05).

Figure 3 Endogenous hormone dynamics in apical leaves of E. giraldii under shading treatments: (A) ABA; (B) IAA; (C) iPA; (D) ZR; (E) GA3.

CK represents the control group, S1 denotes the light shading treatment group, and S2 indicates the moderate shading treatment group. The same letter denotes P > 0.05, and different letters denote P < 0.05.

Endogenous hormone levels in lateral leaves

As shown in Fig. 4, shading stress did not alter the increasing trends of ABA, IAA, iPA, ZR, and GA3 in lateral leaves of E. giraldii, but significantly affected their incremental changes. On July 15, ABA and iPA levels exhibited the pattern: moderate shading group > control group > light shading group, which shifted to moderate shading group > light shading group > control group by July 31 (Figs. 4A, 4C). For IAA content on July 15, the ranking was control group > moderate shading group > light shading group, but reversed to light shading group > moderate shading group > control group on July 31 (Fig. 4B). ZR levels on July 15 showed control group > moderate shading group > light shading group, transitioning to moderate shading group > light shading group > control group by July 31 (Fig. 4D). Similarly, GA3 content on July 15 followed control group > light shading group > moderate shading group, while on July 31 the order became moderate shading group > light shading group > control group (Fig. 4E). No significant differences were observed among all groups (P > 0.05).

Figure 4 Endogenous hormone variations in lateral leaves of E. giraldii under shading treatments: (A) ABA; (B) IAA; (C) iPA; (D) ZR; (E) GA3.

CK represents the control group, S1 denotes the light shading treatment group, and S2 indicates the moderate shading treatment group. The same letter denotes P > 0.05, and different letters denote P < 0.05.

Hormonal balance in apical leaves

Shading treatments altered the trends of ABA/iPA accumulation and affected incremental changes in ABA/IAA, ABA/ZR, and ABA/GA3 ratios in apical leaves (Table 1). On July 15, the ABA/IAA ratio followed: light shading > moderate shading > control, shifting to light shading > control > moderate shading by July 31. The ABA/iPA ratio exhibited light shading > control > moderate shading on July 15, but reversed to control > light shading > moderate shading on July 31. For ABA/ZR, the order was moderate shading > control > light shading initially, transitioning to light shading > control > moderate shading at the later date. Similarly, the ABA/GA3 ratio showed moderate shading > control > light shading on July 15, then changed to control > light shading > moderate shading by July 31. No significant differences were observed among all groups (P > 0.05).

Table 1 Hormonal balance alterations in E. giraldii apical leaves.

	ABA/IAA	ABA/iPA	ABA/ZR	ABA/GA3	
	July 15	July 31	July 15	July 31	July 15	July 31	July 15	July 31	
CK	6.20 ± 0.75a	7.82 ± 3.08a	7.41 ± 0.96a	7.43 ± 1.92a	36.53 ± 8.69a	52.59 ± 9.97a	0.74 ± 0.14a	1.09 ± 0.26a	
S1	7.50 ± 1.79a	9.16 ± 3.64a	7.67 ± 0.32a	7.17 ± 2.21a	36.46 ± 8.12a	56.93 ± 18.22a	0.73 ± 0.06a	0.90 ± 0.32a	
S2	7.32 ± 0.92a	7.73 ± 1.55a	7.12 ± 1.73a	5.31 ± 1.33a	37.35 ± 7.66a	38.81 ± 14.11a	0.75 ± 0.10a	0.81 ± 0.37a	
Note:

CK represents the control group, S1 denotes the light shading treatment group, and S2 indicates the moderate shading treatment group. The same letter denotes P > 0.05, and different letters denote P < 0.05.

Hormonal balance in lateral leaves

Shading stress did not modify the increasing trends of ABA/IAA, ABA/iPA, ABA/ZR and ABA/GA3 ratios in lateral leaves of E. giraldii, but significantly altered their incremental patterns (Table 2). On July 15, the ABA/IAA ratio showed light shading > moderate shading > control, which reversed to control > moderate shading > light shading by July 31. Similarly, the ABA/iPA ratio displayed light shading > control > moderate shading on July 15, but transitioned to moderate shading > control > light shading on July 31. Notably, the ABA/ZR ratio maintained a consistent pattern of moderate shading > light shading > control on both July 15 and 31. For ABA/GA3, the ratio followed moderate shading > light shading > control on July 15, then shifted to control > light shading > moderate shading by July 31. No significant differences were observed among all groups (P > 0.05).

Table 2 Hormonal balance alterations in E. giraldii lateral leaves.

	ABA/IAA	ABA/iPA	ABA/ZR	ABA/GA3	
	July 15	July 31	July 15	July 31	July 15	July 31	July 15	July 31	
CK	5.53 ± 1.71a	8.37 ± 3.17a	6.75 ± 0.60a	7.20 ± 2.16a	31.09 ± 7.23a	44.92 ± 6.13a	0.59 ± 0.07a	1.03 ± 0.28a	
S1	6.72 ± 1.75a	7.30 ± 2.02a	6.80 ± 0.94a	6.82 ± 1.13a	31.24 ± 4.78a	45.69 ± 10.49a	0.60 ± 0.07a	0.94 ± 0.19a	
S2	6.13 ± 1.03a	8.18 ± 2.44a	6.52 ± 1.08a	7.37 ± 3.25a	32.91 ± 5.09a	45.93 ± 11.58a	0.66 ± 0.06a	0.86 ± 0.27a	
Note:

CK represents the control group, S1 denotes the light shading treatment group, and S2 indicates the moderate shading treatment group. The same letter denotes P > 0.05, and different letters denote P < 0.05.

Correlation analysis

As shown in Fig. 5, positive correlations were observed among all measured phytohormones (ABA, IAA, iPA, ZR, and GA3) in both apical and lateral leaves of E. giraldii.

Figure 5 Correlation matrix of phytohormones in apical and lateral leaves of E. giraldii.

AL denotes the apical leaf, and LL refers to the lateral leaf.

Discussion

Plant endogenous hormones are trace organic compounds synthesized within plant tissues that can translocate from production sites to target tissues, capable of triggering physiological responses at low concentrations (<1 μmol/L). These hormones interact through synergistic or antagonistic relationships to coordinately regulate entire growth processes (Anfang & Shani, 2021). The principal categories include IAA, cytokinins, GA3, and ABA. IAA: promotes stem segment elongation, cell division and differentiation, and nutrient translocation (Luo, Zhou & Zhang, 2018). Cytokinins (e.g., iPA and ZR): stimulate cell division, bud differentiation, lateral bud development (thereby breaking apical dominance), and enhance stress resistance (Svolacchia & Sabatini, 2023). ABA: induces dormancy, triggers stomatal closure, inhibits growth, promotes abscission, and improves stress tolerance (Chen et al., 2020). GA3: Mediates stem elongation, induces flowering, breaks dormancy, promotes male flower formation, and delays leaf senescence (Quesada, 2021, 2022).

Analysis of hormonal responses to shading stress and growth strategies in E. giraldii

Our study revealed that E. giraldii exhibits characteristic hormonal network remodeling in response to shading stress. The observed hormonal fluctuations demonstrate that shading induced widespread activation of hormone biosynthesis (ABA, iPA, ZR, GA3), exerting a general upregulatory effect on hormonal levels while simultaneously intensifying organ-specific IAA responses in both apical and lateral leaves (Qiu et al., 2017; Guo et al., 2018). Notably, moderate shading intensity exerted stronger promoting effects on the biosynthesis of iPA, ZR, and GA3 in apical leaves, whereas light shading predominantly enhanced IAA production in lateral leaves. Consistent with our findings, previous research has demonstrated that shading significantly increases IAA, ZR, and GA content in herbaceous peony (Paeonia lactiflora) leaves (Xie et al., 2023), revealing similar phytohormonal response patterns across species.

Analysis of hormonal ratios indicates distinct shade adaptation strategies in E. giraldii: Under moderate shading, apical leaves exhibit relative dominance of growth-promoting hormones (IAA, iPA, ZR, GA3), which likely enhances cell division and elongation to improve photosynthetic capacity in low-light conditions, constituting a “growth-promoting” response. Concurrently, lateral leaves show relative suppression of cytokinins (iPA, ZR) with GA3 and IAA predominance, potentially reducing meristematic activity while promoting leaf expansion, thereby establishing a “limited expansion” pattern. Under light shading conditions, apical leaves demonstrate relative inhibition of IAA and ZR with increased iPA and GA3 levels, which may attenuate apical dominance and shorten internodes, resulting in a “growth-inhibitory” response. On the other hand, lateral leaves display selective ZR suppression coupled with activation of IAA, iPA, and GA3, potentially facilitating the development of an expanded canopy architecture through regulated lateral branch growth.

Therefore, under moderate shading conditions, E. giraldii likely adopts a conservative strategy characterized by “apical growth promotion and lateral growth limitation”. In contrast, under light shading, the species appears to employ an expansive strategy featuring “apical growth inhibition and lateral moderate expansion”.

Perspectives

E. giraldii is a characteristic Chinese medicinal plant. Beyond its stem bark being used medicinally, the young leaves serve as traditional food of the Qiang ethnic group and have been developed into commercial tea products, demonstrating promising industrialization prospects. Although this study addresses the knowledge gap regarding shading effects on endogenous hormones in cultivated E. giraldii, limitations persist including incomplete experimental coverage, relatively short trial duration, and insufficiently in-depth results analysis. To advance the refinement of cultivation techniques and industrial development, future research will integrate growth parameters (e.g., biomass yield, stem elongation), transcriptome profiling, and mechanistic studies on phytohormonal regulation.

Conclusions

This study reveals the remodeling effects of shading stress on endogenous hormone balance in E. giraldii, and elucidating its growth adaptation strategies under shading stress. Under moderate shading, the plant likely adopts a conservative strategy featuring promoted apical growth with limited lateral expansion, whereas light shading induces an expansive strategy characterized by suppressed apical growth coupled with moderate lateral extension. These findings significantly advance the theoretical foundation for E. giraldii cultivation and provide valuable references for understanding endogenous hormonal dynamics and growth strategies in shrubs under shading stress.

Supplemental Information

Supplemental Information 1 Raw data.

All measurements and statistical analyses underlying the figures in the manuscript.

We sincerely acknowledge Mr. Tiancheng Zhu for his professional field management assistance and Chengdu University of Traditional Chinese Medicine for providing laboratory facilities and experimental materials. We are also grateful to all those who contributed to this research.

Additional Information and Declarations

Competing Interests

The authors declare that they have no competing interests.

Author Contributions

Xu Feng Huang conceived and designed the experiments, performed the experiments, analyzed the data, prepared figures and/or tables, authored or reviewed drafts of the article, and approved the final draft.

Rui Gu conceived and designed the experiments, authored or reviewed drafts of the article, and approved the final draft.

Guo Peng Chen performed the experiments, authored or reviewed drafts of the article, and approved the final draft.

Rong Ding performed the experiments, authored or reviewed drafts of the article, and approved the final draft.

Data Availability

The following information was supplied regarding data availability:

The raw data is available in the Supplemental File.

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
