# Peer review of "Effects of shading stress on endogenous hormone levels in Eleutherococcus giraldii: hormonal dynamics and growth strategy analysis"

_PeerJ, doi:10.7717/peerj.20135_

## Round 0.1 · original submission · Minor Revisions

Abiotic stress is a significant factor that affects plant physiology and morphology. Your research provides valuable insights into the effects of shading stress on Eleutherococcus giraldii. However, several technical details should be addressed to further strengthen the article. I strongly recommend carefully reviewing the reviewers' comments and thoughtfully considering each suggestion. If you disagree with any of them, it would be helpful to provide clear, well-reasoned justifications for your viewpoint.

Reviewer 1 ·

Basic reporting

No comment

Experimental design

No comment

Validity of the findings

No comment

Additional comments

All abbreviations should be clearly explained.

Add more details about statistical analysis, which statistical methods were used? Explain clearly the statistical methods.

Give more data on experimental design.

The discussion section needs to be expanded. Include a detailed analysis of your own results.

·

Basic reporting

The manuscript "Effects of shading stress on endogenous hormone levels in Eleutherococcus giraldii: hormonal dynamics and growth strategy analysis" presents a well-designed study investigating the hormonal responses of E. giraldii to varying shading conditions. The research is methodologically sound, with clear experimental design, appropriate statistical analysis, and valuable insights into plant adaptation strategies. The findings are novel and contribute to both theoretical and applied aspects of plant physiology and medicinal plant cultivation. While the manuscript is strong overall, some revisions are needed to improve clarity, statistical reporting, and contextualization of results.

Experimental design

Specify the shade net material and mesh size.
Clarify why the selected light transmittance levels (29.12%, 39.68%) were chosen. Are they ecologically relevant or based on prior studies?

Validity of the findings

Include p-values or effect sizes Tables 1 and 2.
Address whether the short experimental duration (July 15–31) could influence results. Would longer shading treatments lead to different hormonal trends?

Additional comments

The discussion should better Integrate findings with existing literature on shade responses in other plant species, particularly medicinal shrubs.
How do the observed hormonal changes compare with known adaptive strategies?
Authors did not mention that what shading level (29.12% vs. 39.68%) is optimal for E. giraldii growth in agricultural settings?
Figures 3 and 4 needed descriptive captions to stand alone without referring to the text.
Table 2 includes dates in Chinese ("7月15日").
The manuscript occasionally repeats information (e.g., hormonal trends in results and discussion sections).
Streamline the Results and Discussion sections to avoid redundancy (e.g., hormonal trends are described in both).
Ensure consistent formatting of hormone abbreviations (e.g., sometimes "iPA," other times "IPA").

Reviewer 3 ·

Basic reporting

Dear editor: I believe the article is important and may be of interest to science. I recommend accepting it with major alterations. I did not see the hypotheses of the study, nor did I see which statistical tests were conducted.

Experimental design

I miss more detail on how the experiment was set up, particularly how many days were sunny? Were the plants exposed to sunlight and then subjected to treatment? Based on what is pointed out in figure 2 and paragraphs 107 to 109, I have doubts about the shading treatment. Please provide a better description of it. The experimental design.

Validity of the findings

Additionally, conduct statistical tests to indicate whether there are differences between treatments.
The meetings make sense, I suggest that we expand the discussion based on the reanalysis of the data.

Additional comments

Additionally, conduct statistical tests to indicate whether there are differences between treatments. In line 94, I think this data is not adequate 3,293.3 °C. In line 103, I suggest including the year of the study. In line 131, recommend adding Excel.

---

## Round 0.2 · accepted · Accept

I would like to thank you for accepting the referees' suggestions and improving your article based on their suggestions. Your article is ready to publish. We look forward to your next article.

·

Basic reporting

no comment

Experimental design

no comment

Validity of the findings

no comment

Reviewer 3 ·

Basic reporting

The article received improvements after the first version, I think it is suitable for publication, as it contributes to an important line of knowledge.

Experimental design

After the first version, the article received improvements, as the descriptions of the experimental design were expanded.

Validity of the findings

Solid encounters confirmed with the analyses